# A Systematic Review of Bicycle Motocross: Influence of Physiological, Biomechanical, Physical, and Psychological Indicators on Sport Performance

**DOI:** 10.3390/jfmk10020205

**Published:** 2025-06-02

**Authors:** Boryi A. Becerra-Patiño, Aura Daniela Montenegro-Bonilla, Jorge Olivares-Arancibia, Sam Hernández-Jaña, Rodrigo Yáñez-Sepúlveda, Daniel Rojas-Valverde, Víctor Hernández-Beltrán, José M. Gamonales, José Pino-Ortega, José Francisco López-Gil

**Affiliations:** 1Faculty of Physical Education, National Pedagogical University, Bogotá 480100, Colombia; babecerrap@pedagogica.edu.co (B.A.B.-P.); admontenegrob@upn.edu.co (A.D.M.-B.); 2Programa de Doctorado en Ciencias de la Actividad Física y del Deporte, University of Murcia, 30720 Santiago de la Ribera, Spain; 3AFySE Group, Research in Physical Activity and School Health, School of Physical Education, Faculty of Education, Universidad de las Américas, Santiago 7500975, Chile; jorge.olivares.ar@gmail.com; 4IRyS Group, Physical Education School, Pontificia Universidad Católica de Valparaíso, Valparaíso 2530388, Chile; sam.hernandez.jana@gmail.com; 5Faculty Education and Social Sciences, Universidad Andres Bello, Viña del Mar 2520000, Chile; rodrigo.yanez.s@unab.cl; 6Centro de Investigación y Diagnóstico en Salud y Deporte (CIDISAD), Escuela Ciencias del Movimiento Humano y Calidad de Vida (CIEMHCAVI), Universidad Nacional, Heredia 86-3000, Costa Rica; drojasv@hotmail.com; 7Training Optimization and Sports Performance Research Group (GOERD), Faculty of Sport Science, University of Extremadura, 10005 Cáceres, Spain; vhernandpw@alumnos.unex.es (V.H.-B.); martingamonales@unex.es (J.M.G.); 8Faculty of Education and Psychology, University of Extremadura, 06006 Badajoz, Spain; 9Faculty of Sport Science, University of Murcia, 30100 Murcia, Spain; josepinoortega@um.es; 10School of Medicine, Universidad Espíritu Santo, Samborondón 092301, Ecuador; 11Vicerrectoría de Investigación y Postgrado, Universidad de Los Lagos, Osorno 5290000, Chile

**Keywords:** physical abilities, psychological factors, cycling, profile, sport performance

## Abstract

**Background**: This sport involves the integration of various capabilities and mechanisms, including cognitive, physiological, and biomechanical components, that allow the athlete to perform in competition. However, to date, no systematic review has analyzed the indicators that are decisive for sports performance in Bicycle Motocross (BMX). The objective of this work was to carry out a systematic review of the performance variables in BMX and establish recommendations for researchers and trainers. **Materials and Methods**: The following databases were consulted: PubMed, Scopus, and Web of Science. This systematic review uses the guidelines of the PRISMA statement and the guidelines for performing systematic reviews in sports sciences. The search approach, along with the selection criteria and additional details, were previously noted in the prospective registry (INPLASY202480036). The quality of the evidence was evaluated via the PEDro scale. **Results**: The 21 studies that make up the sample of this systematic review have a total sample of 287 athletes. However, in the studies analyzed, there are five main categories for the study of performance in BMX: (i) physiological profile and BMX and bicarbonate; (ii) BMX and physical characteristics (power, speed, and sprint); (iii) translation and rotational acceleration and systems and implements; (iv) psychological variables; and (v) skills and techniques. **Conclusions**: This systematic review provides convincing evidence regarding the influence of several factors that can determine performance in BMX, including P_max_, cadence, neuromuscular capacity, feedback and cognitive training, accelerometry and video analysis, anaerobic–aerobic relationships, physical conditioning, strength, and speed.

## 1. Introduction

“Bicycle Motocross” (BMX) was confirmed as an Olympic sport for Beijing in 2008, and much research has been carried out [1,2,3,4]. However, to date, no systematic review has analyzed the determining factors of sports performance in BMX. This sport is characterized as a sport modality in which batches of eight runners compete along a route that varies between 200 and 400 m and incorporates a series of jumps, curves, and flat surfaces along its route [5]. According to Umarov [6], one of the characteristics of the BMX bike is its 20-inch wheels, total absence of suspension, and presence of a single gear, which allows them to be maneuverable bicycles with a reduced average weight of approximately 12 kg.

This sport is characterized by the interrelation of various capacities and physiological and biomechanical processes [7], which allow the athlete to perform in competition. In that sense, the studies that have been developed in BMX have evaluated the effects of bicarbonate intake on performance [8,9], the effects of accelerometry [10], the effects of translational and rotational accelerations of the head [11], and the effects of induced alkalosis on competitive performance [12]. Similarly, the effects of “pumping and without pumping” techniques have been studied, revealing that the pumping technique significantly favors the production of speed without an additional requirement for muscle activity [13]. Moreover, it has been confirmed that the “maximal mechanical power output”, “maximal theoretical pedaling race”, and “times in 20 m” are significant determining factors of performance in the BMX sprint [14].

In that sense, the BMX race has three specific phases that combine technical and conditional actions for power production. These phases constitute the first phase of the acceleration product of the starting ramp, which demands high levels of power. The second phase requires the athlete to perform actions without pedaling through various obstacles and then pedaling to accelerate again and reach high speed again [15]. Finally, a third phase of resistance to force occurs where the runner must maintain the peaks of cyclical power to develop the maximum speed possible to finish the test [16]. Thus, the first runner who crosses the line in each round is considered the winner of the race [17]. Each of the competitions is defined by four qualifying rounds, each of which is an elimination phase that allows for the determination of the runners who will advance to the end of the event. In the end, the competition time ranges between 39 s (men) and 44 s (women) [18].

With respect to the time spent by runners and, especially, given the effort required, BMX is considered a powerful sport, with an emphasis on resistance to explosive force [19]. The start is made on a slope, so the acceleration capacity and power are fundamental for the first meters [16,20], thus allowing the runners to descend at the maximum possible speed. The study by Herman et al. [21] revealed that there is a significant correlation between the initial acceleration made by the athlete and the result obtained, whereas other studies have reported that there is an association between the levels of maximum power (P_max_) achieved and performance in the race [22,23]. In contrast, in the study carried out by Rylands et al. [17], a Schoberer Rad Meßtechnik (SRM) power measurement system was used to determine speed production through a comparison of the power achieved by the athletes. Six elite BMX racing cyclists were evaluated for reaching power values (1256 ± 256 W), and these values are very similar to those reported for track cyclists and mountain cyclists [24].

Another study that evaluated the production of downhill mountain biking power reported that the maximum output power reached was 834 ± 129 W, whereas the average power was only 75 ± 16 W. The study by Mateo-March et al. [25] evaluated the power in terms of the specificity of the competition and demonstrated that the P_max_ applied on average by the athlete was 85% of the P_max_ that was reached in the laboratory; these values decreased at the beginning, with values of 73% and 51%, respectively. The analysis of the study developed by Hurst and Atkins [26] revealed that the average power only considered 9% of the maximum values, whereas an average of 38 pedaling actions per race was found, with average pedaling periods of 5 s, which shows that power and cadence do not correlate with the times reached in the race. Therefore, it is necessary to understand that performance in training and competition may vary. In addition, there are differences between laboratory and field-specific performance tests for BMX cycling [27]. There are different instruments, such as potentiometers, that are responsible for evaluating the power used by the athlete in terms of the specificity of the competition [28,29]. In response, portable sensors oversee estimating the production of mechanical power during cycling [30].

Thus, the estimation of power in the real context of competition is necessary because it is a useful and objective variable to monitor resistance in sports of a cyclical nature, such as cycling, which is key here for mechanical power [31], and, on the other hand, it allows for the evaluation of the external load produced in training and/or competitions [32,33]. Similarly, another study developed by Koellner et al. [34] investigated the acceleration and deformation responses of a BMX bicycle in common racing situations and concluded that in race simulations, cutoff stresses can be produced that are 84% higher than those in static tests. In this way, these technological advances have allowed for a better understanding of the specific demands of BMX competitions.

Compared with the above, the need to continue developing studies that analyze the effects of the indicators on sports performance is confirmed because the knowledge acquired is useful to favor a greater understanding of the sports discipline [35] and, with this understanding, to understand the knowledge and findings to be used by coaches and transferred to the real context of sports preparation, especially for competitions [36]. To the best of our knowledge, no systematic review has evaluated the determinants of BMX performance. Therefore, the objective of this research was to carry out a systematic review of the published studies on the performance variables in BMX and establish recommendations for trainers, researchers, and practitioners. In this review, studies focused on the compilation of statistical analyses in search of a better understanding of the relationships between physiological factors and sports performance.

## 2. Materials and Methods

### 2.1. Design

This systematic review uses the guidelines of the PRISMA statement (Preferred Reporting Items for Systematic Reviews and Meta-Analyses) [37,38] and the guidelines for performing systematic reviews in sports sciences [39]. The search approach, along with the selection criteria and additional details, were previously noted in the prospective registry for the International Platform of Registered Systematic Review and Meta-analysis Protocols (INPLASY202480036).

### 2.2. Sources of Information

The search strategies consider the following characteristics.

Date: 1 April 2024.

The following databases were consulted: PubMed (Medline), Scopus, and Web of Science.

### 2.3. Inclusion and Exclusion Criteria

The search was carried out by two authors, who identified the most relevant information of each of the studies. The information (title, authors, date, and database) was then downloaded into an Excel document, where duplicates were selected and deleted. If any relevant study was found that was not found through the search strategy, it was added as “external sources”. For the selection and inclusion of studies in the present work, a series of inclusion and exclusion criteria were established based on the PICO strategy (Table 1).

### 2.4. Search Strategy and Data Extraction

The 21 studies that make up the sample of this systematic review have a total sample of 287 athletes. Specifically, several studies do not refer to the characteristics of the evaluated sample (sex, number of athletes, age, and other analysis variables). To design the search strategy, strategies P (population), I (intervention), C (comparison), and O (outcomes) were applied, as suggested by the guidelines used for this systematic review. The Boolean commands “AND” and “OR” were used to group the mentioned terms. For each of the databases, we proceeded similarly. Before arriving at the construction of the definitive search phrase for each database, the possible combinations were tested with the following list of words: “Sport”, “Cycling”, “Motocross”, “Sports performance”, “Physiological Indicators”, “Physiology”, “Profile”, “Measurements”, “Riders”, “BMX”, “Physiological Factors”, “Power Analysis”, “Performance”, “Field-Based Bicycle Motor Cross”, “Wearables”, “Biomechanical Variables, and “Psychological Factors”. The review of the different searches was carried out by two authors independently to define the terms that yielded the largest number of documents related to the subject. The selected terms were “BMX”, “Elite Riders”, “Physiological Indicators”, “Profile”, “Biomechanical variables”, “Psychological factors”, and “Performance”. Finally, based on these terms, the following search equation was built: (BMX OR riders) AND (“Physiological Indicators” OR profile OR “Biomechanical Variables” OR “Psychological Factors” OR performance). This search chain was adapted for the PubMed, Scopus, and Web of Science databases. The search using controlled vocabulary was carried out with the search by keywords to improve recovery. The searches were carried out to identify studies without other restrictions in reference to the publication date, language, or study design. Similarly, citation searches were carried out for the key studies included. When it was not possible to obtain the full texts of the articles from the institutional or open access subscriptions, an attempt was made to contact the corresponding authors directly.

After the identification of the studies, the documents were downloaded in Excel with the following data for each study: (i) title; (ii) authors; (iii) journal; (iv) year; and (v) database. Once the documents from each database were downloaded, a single database was unified. Likewise, if any document was found that did not appear in the search strategy, it was added through external sources. The process was carried out independently by two of the authors (“A.M.-B.” and “B.A.B.-P.”). Any disagreement (5% of the total documents) on the final state of inclusion–exclusion was resolved through academic discussion, both in the selection phase and in the exclusion phase. In the discussion process, the two independent authors analyzed the articles at the same time following the criteria set out in the order of Table 2. This process was systematized in Excel. Thus, to follow the PRISMA guidelines, data extraction was overseen by “A.M.-B.” and “B.A.B.-P.”.

### 2.5. Methodological Quality

The quality of the evidence of the articles included in this review was evaluated via the PEDro scale [40]. This scale is based on criteria that allow us to identify whether the studies have sufficient internal validity and statistical information to interpret the results (external validity (item 1), internal validity (items 2–9), and statistical information (items 10–11)). Each item was classified as yes or no (1 or 0, respectively), depending on whether the criterion was met in the study. The total score considers items 2 to 11; therefore, the maximum score was 10 [41]. For the quality of the evidence, scores < 4 are considered poor quality, 4–5 are considered moderate quality, 6–8 are considered good, and 9–10 are considered excellent [41]. In this review, 203 items (96.6%) were evaluated through agreement between two reviewers, and the remaining items were evaluated according to the average of the studies (Table 2). The quality of the evidence oscillated between the “moderate-good” categories, as some studies did not present randomization in the selection of the sample, nor did they have a control group. In addition, the quality of the evidence was heterogeneous throughout the studies. Therefore, the quality of the evidence was defined based on consensus of the researchers as “moderate”, which means that there was not good methodological quality [41].
jfmk-10-00205-t002_Table 2Table 2Methodological quality of the studies.
Items
Studies (Authors)1234567891011Total PEDroTSQElvira et al. [10]110100010115ModMoya-Ramón et al. [42]100000010113LowDaneshfar et al. [43]110100010115ModMateo-March et al. [44]100100010114ModRobert et al. [45]100101010115ModDaneshfar et al. [23]101100010115ModPetruolo et al. [46]100100010114ModGross and Gross [47]100000110114ModPeinado et al. [12]110011010116GoodHurst et al. [11]100100110115ModRylands et al. [48]100100011115ModRylands et al. [49]100000010113LowMateo-March et al. [25]100010010114ModLouis et al. [50]100100010114ModRylands et al. [17]100000111115ModMateo et al. [51]100000011114ModCowell et al. [20]100000010113LowZabala et al. [9]110111010117GoodMateo et al. [16]100000010113LowZabala et al. [19]100000010113LowZabala et al. [8]100100010114Mod

### 2.6. Identification and Selection of Studies

In total, 6353 documents were identified. After reviewing the final database initially, documents were deleted because of duplication (*n* = 1681) and because they were not related to the subject (*n* = 4557). Of the 115 screened documents, only 46 were analyzed through a first review of the title and the summary to confirm their inclusion, after which we proceeded to obtain access to the complete study for systematic reading (Figure 1). Thirty-two documents for eligibility were analyzed in their entirety, and eleven of them were excluded. After this process, 21 articles met all of the inclusion criteria. In all of them, the risk of bias was evaluated, and the most representative data of the sample are included in Table 3.

### 2.7. Analysis of the Participants

The 21 studies that make up the sample of this systematic review include 287 athletes, 13 of which evaluate men, 3 of which consider women, and 2 of which consider men and women. This fact is included in Table 4, which specifies the characteristics of the sample selected in each of the studies, to contextualize this information. Specifically, several studies do not refer to the characteristics of the evaluated sample (sex, number of athletes, age, and other analysis variables).

## 3. Results and Discussion

The outcome measures included in the current systematic review were those evaluated by at least 2 of the 21 articles. These variables were (i) BMX and physiological characteristics; (ii) BMX and physical characteristics (strength, power, speed); (iii) biomechanical and technological; (iv) psychological variables; and (v) skills and techniques.

### 3.1. What Are the Most Important Physiological Variables?

#### 3.1.1. Physiological Profile

Three studies were found that considered BMX and physiological profile categories, with a total of 34 participants. In this sense, Petruolo et al. [46] expressed various results that reflect the following physiological demands: a maximum oxygen consumption (Vo2max) of 55.7 ± 4.8 mL/min/kg, with a P_max_ during a short sprint on a bicycle of 1498 ± 189 W, and an average power during the Wingate test (WT) of 1344 ± 158 W. In addition, peaks were recorded in the vertical jump of 58.6 ± 7.7 cm, 4625 ± 768 W, and 64.3 ± 7.5 N/kg. In addition, in the simulation of the BMX race, a slight improvement in performance time was observed, accompanied by a significant increase in the perception of effort, as well as a significant increase in the concentrations of lactate and hydrogen ions in the blood throughout the races, whereas the concentrations of bicarbonate decreased (*p* < 0.001) [46]. These results highlight the high anaerobic demand and intensity of BMX competitions, as well as the importance of muscle power and endurance capacity in cyclists.

In previous research by Louis et al. [50], simulated competition was shown to induce a high request for both aerobic (Vo2 maximum average (Vo2peak): 94.3 ± 1.2% Vo2max) and anaerobic (average blood lactate: 14.5 ± 4.5 mmol × L^−1^) glycolysis during each race. In addition, the repetition of the six cyclists separated by 30 min of recovery caused a significant deterioration in the acid–base balance from the third race to the sixth race (average decrease in excess base content: −18.8 ± 7.5%, *p* < 0.05). Finally, the most recent study by Daneshfar et al. [43] revealed that (a) the BMX return time was significantly correlated with the mean PWR, but the strength of this association decreased as successive turns were performed; (b) the subjects demonstrated a high contribution of aerobic metabolism during the laps and showed a significant correlation with the mean lap times; this association indicated an incremental trend; and (c) the blood lactate response (BLr) media correlated significantly with the half lap time, and the correlation between BLr and the time for each lap was stronger in the last laps. According to the results, despite the short cycling time (~35 s) on each lap of BMX, both aerobic and anaerobic energy systems were associated with performance.

#### 3.1.2. BMX and Bicarbonate

In an investigation developed by Zabala et al. [8], the impact of sodium bicarbonate intake (NaHCO_3_) on performance and physiological responses was evaluated during three WT tests and compared to that of a placebo. No significant differences were found in peak power (PP) or time to peak power (TPP) between NaHCO_3_ and placebo conditions in any of the three WTs, with PP values of 1607 ± 310 W and 1599 ± 265 W, respectively, in the first WT. The average power (MP) and the fatigue index (FI) also showed similar results between the two conditions. Positive correlations were observed (*r* = 0.40–0.87, *p* < 0.05) between the PP reached under NaHCO_3_ and the control samples. In terms of the countermotion jump (CMJ), there was a significant improvement after the first WT in the NaHCO_3_ condition compared with the control (36 ± 1.3 cm vs. 34.1 ± 1.3 cm, *p* = 0.027). However, no significant differences were observed after the second and third WTs, although the performance tended to be better with NaHCO_3_ (average improvement of 1.44 cm, ~4%). For blood lactate (BLa), no significant differences in levels were found at 30 min after each WT between the two conditions. However, a relevant increase in [BLa] was observed 3 min after the second WT in the NaHCO_3_ condition (*p* = 0.014), and [BLa] tended to increase after the third WT (*p* = 0.07). In terms of the perception of effort (RPE), there were no significant differences between the tests for any of the RPE scales, although the CR10 scale tended to differ between the two conditions (first WT, *p* = 0.06; third WT, *p* = 0.09). The perceived preparation was significantly greater for the second WT and the third WT under NaHCO_3_ conditions (*p* = 0.023 and *p* = 0.019, respectively). Eight of the nine cyclists felt that their best performance occurred under the effect of NaHCO_3_.

Similarly, Zabala et al. [9] investigated the effects of the ingestion of sodium bicarbonate in elite BMX cyclists and reported no significant differences between bicarbonate and placebo in terms of PP (1.610 ± 373 vs. 1.599 ± 370 W), average power (MP) (809 ± 113 vs. 812 ± 108 W), TPP, or the fatigue index. Regarding the perception of effort, no significant differences were found in the Effort Perception Rating (RPE) 6–20, CRIO RPE, or Readiness Perception (Scale 1–5). With respect to acid–base equilibrium, significant differences were observed in pH before the first test (7.47 ± 0.05 with bicarbonate vs. 7.41 ± 0.03 with placebo) and in bicarbonate concentration [HCO_3_^−^] before the first test (29.08 ± 2.27 mmol L^−1^ with bicarbonate vs. 22.85 ± 0.24 mmol L^−1^ with placebo). Despite changes in the acid–base balance, the ingestion of baking soda did not improve performance or perception of effort in elite BMX cyclists throughout consecutive tests.

Finally, in a more recent study, Peinado et al. [12] reported significant differences in several physiological variables between the conditions of sodium bicarbonate (NaHCO_3_^−^) and placebo. NaHCO_3_^−^ increased blood pH (*F*(1, 22.01) = 79.15, *p* < 0.001), bicarbonate (*F*(1, 19.86) = 12.76, *p* = 0.002), and excess base (*F*(1, 17.50) = 38.45, *p* < 0.001). There were no significant differences in lactate levels (*F*(1, 15.22) = 3.19, *p* = 0.09). The performance variables (time, maximum speed, and time to maximum speed) and heart rate (average and maximum) did not significantly differ between the conditions. However, the effect of race was significant for the mean (*F*(2, 44.49) = 3.46, *p* = 0.04) and maximum (*F*(2, 43.38) = 11.07, *p* < 0.001) heart rates. Regarding the perception of effort, there were no significant differences between conditions, although race influenced scales 1–5 (*F*(2, 43.79) = 3.34, *p* = 0.04) and RPE 6–20 (*F*(2, 42.93) = 3.69, *p* = 0.03). NaHCO_3_^−^ increased heart rate variability (*F*(1, 69.56) = 5.05, *p* = 0.02). No significant effects of the condition were observed on Vo2, VCo2, or VE, although there were significant differences between pre- and post-measurements (*p* < 0.001). These results suggest that NaHCO_3_^−^ improves some aspects of acid–base balance and heart rate variability, but it does not significantly affect direct physical performance or exercise perception compared with the placebo.

### 3.2. Is There a Relationship Between Physical Capabilities and BMX Performance?

#### BMX and Physical Capabilities (Power, Speed and Sprinting)

In this category, different studies were found, such as that of Mateo-March et al. [16], where in free pedaling races (FPs) the registered PP was 1.144.67 ± 28.65 W, corresponding to 85.21 ± 2.15% of the P_max_ of 1.343.12 ± 68.63 W, which reached 1.42 ± 0.02 s. The PP decreased to 73.02 ± 18.38% of P_max_ during the exit from the door and 51.37 ± 15.84% in the first straight path. A total of 88.76% of the race time was spent on power production below 50% P_max_, with only 3.23% of the time above 50% P_max_ on high-difficulty (HD) tracks. The average (MV) and peak (PV) speeds were higher in the FP condition (34.21 ± 1.00 km/h and 43.09 ± 1.31 km/h), followed by the sprint from the door (GSP) and no pedaling (NP) conditions. Under the NP condition, the MV was greater on the HD tracks (29.00 ± 0.89 km/h), and the highest PV occurred on the LD tracks (36.05 ± 1.41 km/h). In GSP, the MV was greater in HD (30.17 ± 1.40 km/h), and the highest PV was observed in LD (39.19 ± 1.39 km/h). In the FP, the highest MV and PV rates corresponded to the LD tracks (35.63 ± 1.07 km/h and 44.35 ± 1.18 km/h, respectively). Multivariate ANOVA revealed significant effects of difficulty (*F* = 5.55; *p* = 0.013), technique (*F* = 390; *p* = 0.001), and the interaction between difficulty and technique (*F* = 53.9; *p* = 0.001). Spearman correlations between MP and MV in FPs were regular for HDs (*p* = 0.02; *r* = 0.45) and LDs (*p* = 0.01; *r* = 0.61). In summary, for BMX cyclists, using a free pedaling technique and running on less difficult tracks maximize power and speed during the race.

Continuing with these findings, the results of the study by Rylands et al. [17] revealed that BMX cyclists had low levels of P_max_ during 50 m performance tests, with a combined average value of 1284.5 ± 254.5 W. In the 200 m fatigue tests, maximum and minimum powers of 1256 ± 276 W and 450 ± 135 W, respectively, were observed, along with power fatigue rates of 57 ± 3.98 W·s^−1^ and power fatigue scores of 64.77 ± 3.01%. In addition, the cadence and speed increased as the power decreased during the fatigue tests. These findings highlight the dynamic relationships among power, cadence, and speed in BMX cyclists during prolonged sprint efforts. Similarly, Rylands et al. [49] investigated the validity of the P_max_, torque and power production time tests in elite BMX cyclists, comparing the results obtained in the laboratory and the field. The results revealed significant differences between the laboratory and field tests for P_max_ (*t*(7) = −11.38, *p* < 0.01) and power production time (t(7) = 8.64, *p* < 0.01), with correlation coefficients of *r* = 0.78 and *r* = 0.86, respectively. With respect to torque, no significant differences were found between the laboratory and field tests (t(7) = −1.48, *p* = 0.18, *r* = 0.61). The figures show higher agreement limits for P_max_ in field tests (246–714 W) and shorter times for P_max_ production in the field than in the laboratory (−3.39 to −0.73 s). The agreement limits for the torque were 32.99 to 19.07, indicating slightly higher values in the laboratory, although the difference was not statistically significant.

Rylands et al. [48] subsequently investigated the effects of cadence selection on P_max_ and the time to reach P_max_ on elite BMX cyclists. The analysis did not reveal significant differences in P_max_ (*p* = 0.424) or time to reach P_max_ (*p* = 0.532) among the cadences of 80, 100, 120, and 140 rev/min. However, the optimal cadence for the time to reach P_max_ was 120 rev/min (2.50 ± 1.07 s), whereas cadences of 80 and 140 rev/min resulted in longer times (3.50 ± 0.88 s and 3.50 ± 0.80 s, respectively). The optimal cadence for P_max_ was 100 rev/min (1105 ± 139 W), with a 140 rev/min cadence resulting in the lowest P_max_ (1046 ± 175 W). Large effect sizes were observed when comparing the time to reach P_max_ and P_max_ between the different cadences, highlighting the significant influence of the selected cadence on the performance of the cyclists.

Moreover, on a more current date, Gross and Gross [47] developed a study in which 12 BMX cyclists who completed jump tests and sprints participated. The findings show that the force–speed (Fv) and torque–cadence (Tc) profiles are strongly associated (coefficient of determination [*R*^2^] of 0.92 for Fv and between 0.91 and 0.95 for Tc). The results indicated typical errors in the Fv and Tc characteristics, with ranges of 1.4–2.6% and 2.3–5.1%, respectively. The tests revealed that the parameters F0 and P_max_ of the jump were strongly correlated with the parameters for the average torque (Tor) and P_max_ of the ramp start (*r* = 0.73–0.99), although the correlations between speed and cadence, as well as between SFv and STc, were low. When ramp starts were compared with flat sprints, very high correlations were observed (Pearson correlation coefficient [*r*] = 0.86–0.97), and there were significant differences in P_max_ (*p* = 0.01), highlighting greater power in ramp starts. The dichotomous classification of the participants revealed divergences in the correlations of Tor and strength in 4 of 11 cyclists and in cadence and speed in 7 of 11 cyclists, which suggests variability in adaptation to different test conditions.

Research by Daneshfar et al. [23] shows that postrace analysis of power data could help cyclists recognize the need to apply certain strategies in relation to pedaling rates and energy production in certain parts of the BMX track, especially at the beginning and around the first corner. BMX trainers should consider designing training programs based on race intensity and power output zones. The race time was also significantly associated with the time curves (*r* = 0.58, *p* < 0.001). P_max_ (1288.7 ± 62.6 W) was reached in the first 2.34 s of the race. With zero values included, the average power was 355.8 ± 25.4 W, approximately 28% of the P_max_, whereas it was 62% when zero values were excluded (795.6 ± 63.5 W).

In addition, Robert et al. [45] verified the existence of an indirect relationship between the time spent completing the circuit and the jump height reached in the jumps of SJ (*r* = –0.801; *p* = 0.17), CMJ (*r* = –0.798; *p* = 0.18), and DJ (*r* = –0.782; *p* = 0.22), which suggests that developing the different strengths through specific training of concentric capacity (SJ), explosive elastic (CMJ), and explosive elastic reflex (DJ) could have a direct relationship with the times obtained in the race. The results of this study suggest the existence of a direct relationship between the best mark obtained in BMX and the jump capacity of the SJ, CMJ, and DJ tests. The absolute values in the form of flight height for SJ, CMJ, and DJ jumps were greater in the GE. No significant differences were found for the variables IE and IF, even though the absolute values of GE were greater than those of GR. Finally, a more recent study by Moya-Ramón et al. [42] described the sprint characteristics of elite BMX cyclists and whether these variables predict performance in this population. Pearson correlation and linear regression analysis revealed a significant association (*p* < 0.05) between the field performance of cyclists and the 1RM (*r* = 0.84), P_max_ production in the RST (*r* = 0.87), and P_max_, the average power, and lactate production in the WT (*r* = 0.680), as did the P_max_ of the force speed profile (*r* = 0.71), which translates into maximum power production (PPO), which is a key predictor of performance in this discipline, explaining 78% of the variability in full return performance. Although other measurements, such as 1RM, PPwing, P30s, lactate, and P_max_, also show significant correlations with performance, PPO stands out as the main factor.

Overall, these studies highlight the importance of various strategies and conditions in the performance of BMX cyclists. Training on less difficult tracks and using free pedaling techniques maximize power and speed during races. Proper cadence and power management are crucial to maintaining performance in prolonged sprints, with an optimal cadence between 100 and 120 rev/min. Real-world evaluations are more indicative of performance than laboratory tests. In addition, training on flat surfaces and ramps improves power under different conditions, and focusing training on the start and first curves of the track is essential for optimizing race times. Finally, the development of strength through specific jumps directly correlates with better performance in races, highlighting the need for comprehensive training programs that address all of these areas.

### 3.3. What Are the Most Studied Biomechanical and Technological Variables Related to BMX Sport Performance?

These last results are composed of other performance variables identified in the research process; thus, in this sports discipline, the following was found.

#### 3.3.1. Translation and Rotational Acceleration

The work of Hurst et al. [11] revealed that the range of motion of the cervical spine significantly differed depending on age, specifically in the conditions of flexion and extension. In terms of cervical flexion, significant differences were found between the groups aged 6–9 years and 10–13 years (*p* = 0.005) and between the groups aged 6–9 years and 14–18 years (*p* = 0.003). There were no significant differences between the groups aged 10–13 years and those aged 14–18 years. Similarly, when cervical extension was analyzed, significant differences were observed between the groups aged 6–9 years and those aged 14–18 years (*p* = 0.02). No significant differences were found between the other age groups or conditions. These results indicate that younger children (6–9 years) have a different range of cervical movement than older adolescents do, especially in terms of flexion and extension.

With respect to repetitive head movement (RHM) and acceleration, significant differences were found by age group (*F*(2.23) = 26.76; *p* < 0.001; ηp2 = 0.73). On the other hand, the number of accelerations did not significantly interact with condition or age, but there were significant main effects for condition (*F*(1.20) = 6.00; *p* = 0.02; ηp2 = 0.23) and age (*F*(2.20) = 3.51; *p* = 0.04; ηp2= 0.26). However, post hoc comparisons did not reveal significant differences between individual age groups [11].

Another variable of the study developed by Hurst et al. [11] was translational acceleration, where no significant effects of interaction or main-by-condition effects were found, but a significant main effect for age was found (*F*(2.20) = 5.55; *p* = 0.01; ηp2 = 0.36). Post hoc comparisons revealed significant differences between the groups aged 6–9 years and 14–18 years (*p* = 0.04) and between the groups aged 10–13 years and 14–18 years (*p* = 0.02). Finally, with respect to rotational acceleration, a significant effect of age was not found, but there was a significant main effect of condition (*F*(1.20) = 7.15; *p* = 0.02; ηp2 = 0.26). A significant relationship was found between RHM and age (*r* = 0.83; *p* = 0.001). There was also a significant relationship between the RHM and the number of accelerations in the no-load condition (NB) (r = 0.46; *p* = 0.03).

These results suggest that the range of cervical movement and dynamic behavior of the head vary considerably with age in children and adolescents. Compared with older adolescents (14–18 years), younger children (6–9 years) present more notable differences in cervical flexion and extension, possibly indicating less cervical stability or control in the youngest children. Translational accelerations are more affected by age, whereas rotational accelerations are more influenced by experimental conditions. The significant relationships between RHM and age, as well as with the number of accelerations in the condition without load, suggest that both physiological development and external conditions influence head movement in this population [11].

#### 3.3.2. Systems and Implements

In relation to another aspect, three studies were found that discuss various implements (plates and rings) and if they influence performance. One of these studies was that of Elvira et al. [10], who described the acceleration profile in the slope departure of BMX cycling and compared not-Q-ring chainring (NQ) and Q-ring chainring plates (Q). Therefore, discrete-time, acceleration, and statistical parametric mapping (SPM) were used to compare the conditions. Q did not improve performance, despite an increase in force application time (*p* = 0.013, ES = 0.39) and a reduction in dead point time (*p* = 0.028, ES = −0.73). The time after the four pedal stroke (PS) events was longer (*p* = 0.006, ES = 0.63), and the time after 3 m did not change. Similarly, the statistical parametric mapping (SPM) 1D comparison revealed no differences across the four PS events. The combined methodology (accelerometry and video analysis) has proven useful in characterizing the cyclist’s ability to apply force in the first pedal strokes in BMX cycling, differentiating between positive and negative acceleration phases. On the other hand, the results of this study show that, in practice, the QR noncircular pedaling system was not able to improve the overall performance on the BMX output slope. In addition, these results show that the Q-ring system does not seem to generate mechanical changes that can improve pedaling performance, nor does it increase the amount of time cyclists accelerate while starting climbing beyond the first pedal.

Mateo-March et al. [25] investigated the effect of noncircular chainrings (Q-rings) on performance during the initial acceleration phase of a BMX race and reported that the elite group covered a greater distance via the Q-ring (+0.26 m, *p* = 0.02; d = 0.23), whereas the improvement for the cadet group (+0.04 m) was not significant (*p* = 0.87; d = −0.02). In addition, there was no significant difference in output power for the elite group, whereas the cadet group revealed a higher P_max_ with circular chaining. Neither the lactate level nor the heart rate significantly differed according to the different chains used. This means that the best BMX cyclists were able to significantly improve performance (total distance) in the initial sprint with the Q-ring system, despite the absence of significant differences in power outputs. Increasing the total distance traveled in the first 3.95 s at 0.26 m means an improvement of 1.12% and may be a sufficient reason to favor the chaining of the Q-ring, as it could translate into several meters at the finish line.

Mateo-March et al. [44] compared determinants of pedaling performance in the first meters of the acceleration phase in BMX, with two different CR systems, circular and noncircular, in elite and cadet cyclists. In the elite category group, no differences were found between the circular and noncircular CRs in total distance, reaction time, or %PM. In the cadet group, differences were found between the circular and noncircular CR conditions in %PM (*p* = 0.02), resulting in better results for the circular CR. In addition, no differences were found in total distance or reaction time. This means that the use of the noncircular CR system does not improve the performance of either elite or cadet cyclists.

### 3.4. How Do Psychological Variables Influence BMX Performance?

Another performance variable in BMX is the emotional aspect, which requires more study; however, the study by Mateo-March et al. [51] analyzed how BMX competition affects the subjective perception of anxiety and whether this emotional alteration is reflected in the dynamics of CF. For this reason, a main effect of the test day was found in the components of cognitive anxiety (AC), somatic anxiety (SA), and self-confidence (SC), as well as the correlation between SA and AC on the first day of competition (*r* = 0.774; *p* < 0.05), which decreased on the second day. With respect to heart rate variability (HRV), significant differences in several indices, including the mean RR interval (RRi), the standard deviation from normal RR intervals (SDNN), the mean quadratic difference of successive normal R-R intervals (rMSSD), the high-frequency power of R-R intervals (lnHF), the low-frequency and high-frequency power ratio (LF/HF) (*p* < 0.01), and the low-frequency power of R-R intervals (lnLF=) (*p* < 0.05), were detected between the resting (r) and anxiety (a) conditions. In addition, significant differences were found between the measurements of the two days of competition in RRi, SDNN, rMSSD, lnHF, LF/HF, and sample entropy (SampEn) (*p* < 0.01) and in lnLF and LF/HF (*p* < 0.05).

With respect to the scores of Competitive State Anxiety Inventory-2 Revised (CSAI-2R) and the dynamics of the HR, a positive correlation was found between a1 and SA on the first day of the competition (r = 0.72; *p* < 0.05), where SA explained 51.8% of the variability of a1. However, this relationship was not maintained on the second day. In summary, significant changes in precompetitive anxiety and HRV were observed during the competition, as were some associations between anxiety scores and certain HR rates [51].

The findings of the study by Mateo-March et al. [51] indicate that precompetitive anxiety during BMX competitions significantly affects the dynamics of heart rhythm, resulting in a decrease in vagal control and an increase in sympathetic activity. This change in the balance of the autonomic nervous system is reflected in a reduction in the temporal and spectral rates of HRV, along with an increase in the LF/HF ratio. In addition, the analysis suggests that the perception of anxiety correlates with specific patterns in the dynamics of heart rhythm, with higher levels of perceived anxiety associated with a more regular heart rhythm pattern, indicating vagal withdrawal. In addition, repeated exposure to competitive stress can attenuate the subjective perception of anxiety over time.

### 3.5. Skills and Techniques in BMX?

With respect to the last two variables, Zabala et al. [19] determined the effect of external feedback management (FB) on the time used to execute the door initiation ability. The results revealed that there were no significant differences between the first two pretest sessions (PRE) or between any of the other treatment, posttest, or retest sessions (TREAT, POS, and RET, respectively). However, significant differences were observed between all of the PRE sessions and all of the TREAT, POS (posttest), and RET sessions (*p* < 0.028), indicating a significant reduction in the time required to perform this skill after TREAT (1.264 ± 0.045 ms in PRE, 1.047 ± 0.019 ms in POS, and 1.041 ± 0.021 ms in RET). This means that the use of audiovisual FB and cognitive skill training can result in a significant improvement in the execution of the start of the door in BMX, reducing the time to develop the task.

The last variable, which was developed by Cowell et al. [20], determines that BMX competitions involve a unique combination of skills and techniques, with each section of the track presenting specific demands. Therefore, for each race, the movement patterns and the time spent pedaling, jumping, and “pumping” were determined. On average, elite men took 39.62 ± 0.78 s to complete a track, used 30.45 ± 3.2 pedals, and spent 11.83 ± 1.11, 9.64 ± 1.79, and 17.05 ± 1.51 s pedaling, jumping, and “pumping”, respectively. Elite women took 40.95 ± 0.91 s to complete a track, used 33.65 ± 5.06 pedals, and spent 14.40 ± 2.17, 6.28 ± 1.41, and 17.80 ± 1.83 s pedaling, jumping, and “slipping and pumping”, respectively. The dominant movement patterns investigated for the start, take-off, landing, and pumping were hip extension (~30 times per turn), knee extension (~30 per leg per turn), and abduction and horizontal adduction of the shoulder (20 times per turn). This means that BMX competitions involve a unique combination of skills and techniques, with each section of the track presenting specific demands. The initial start and execution are critical to the overall success of the race, establishing a movement pattern that influences performance in the following sections.

#### 3.5.1. Limitations

The main limitations focus on the number of documents found and the diversity of factors used to evaluate the different studies. Notably, no longitudinal studies have investigated how certain variables influence the performance of BMX athletes during the development of a full season (training and competitions). Another difficulty encountered is that several studies did not refer to the characteristics of the population, which made it difficult to interpret the results. This converges in the difficulty of analyzing the comparative measures between the different investigations analyzed in this review.

The studies included in this systematic review, which were too heterogeneous, did not allow for a meta-analysis, and although this type of study, which was carried out to analyze BMX, does not allow for solid conclusions to be reached, the information contained in Table 4 reflects important information from each study individually.

#### 3.5.2. Future Recommendations

The main recommendations are associated with possible research questions that are still unresolved and motivated by the low number of documents related to the study of variables and performance indicators in BMX. First, future lines of research should consider the development of longitudinal and experimental studies that allow for the establishment of causal relationships and correlations that allow for the determination of the interactions among physiological, biomechanical, physical, technical, tactical, cognitive, psychological, and other skills, as well as multicomponent studies. On the other hand, careful review of the use of laboratory tests to incorporate these advances in science into the practical work of coaches and athletes is suggested. A third approach can focus on the study of BMX in children and youth categories, with the goal of understanding developments and improvements in performance in response to the evaluated capacities.

## 4. Conclusions

This systematic review provides convincing evidence regarding the influence of several factors that can determine performance in BMX, including P_max_, cadence, neuromuscular capacity, feedback and cognitive training, accelerometry and video analysis, anaerobic–aerobic relationships, physical conditioning, strength, and speed. All of these qualities must be integrated into the techniques of the sport, including pedaling, jumping, landing, and pumping. Moreover, factors like the use of supplements, including sodium bicarbonate, do not help improve performance in BMX. However, more scientific evidence is needed to continue understanding how other emotional, cognitive, and psychological variables regulate this process and influence sports performance in terms of age, sports and competitive experience, gender, and the practice of BMX.

The results of this systematic review revealed that the performance factors in BMX are associated with the physiological processes that prevail over the psychological, technical, and strategic processes, with the production of power in the different phases of the race being one of the most studied indicators. The findings of this review should be incorporated with caution by trainers and interdisciplinary teams to design specific training programs associated with the power phases expressed by each athlete. In contrast, various studies have shown that laboratory tests may not consider the true performance capacity athletes can exhibit in the field.

## Figures and Tables

**Figure 1 jfmk-10-00205-f001:**
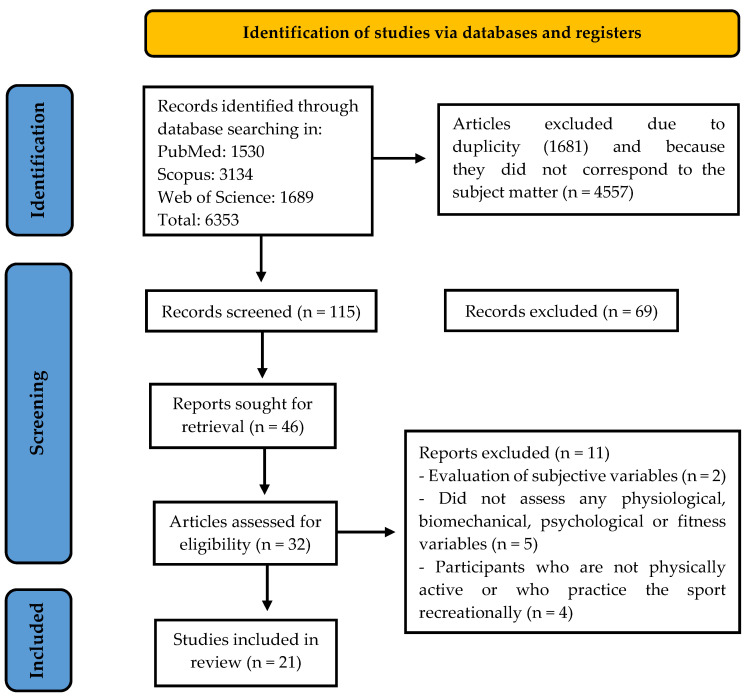
Flow diagram of the systematic review.

**Table 1 jfmk-10-00205-t001:** Inclusion and exclusion criteria.

Topic	Inclusion	Exclusion	Search Keywords
**Population**	Be athletes	Participants who are not physically active	(BMX OR riders)
**Intervention**	Evaluation of at least one physiological, biomechanical, or physical capacity or psychological variables in training and/or a competition variable	Focused only on the evaluation of subjective variables	
**Comparison and outcomes**	Results that relate physiological indicators (for example, power), profiles of physical abilities (for example, strength, endurance), biomechanical factors (accelerations, angles, etc.), and psychological (feedback) and technical skills	Results that do not relate physiological indicators (for example, power), profiles of physical capacities (for example, strength, resistance), biomechanical factors (accelerations, angles, etc.), or psychological (feedback) and technical skills	(“Physiological indicators” OR profile OR “Biomechanical variables” OR “Psychological factors” OR performance)
**Study design**	-	-	-
**Other criteria**	- Original research that has been reviewed by academic peers	- Systematic reviews, meta-analysis, bibliometric analysis, narrative or literary reviews - Patents, abstracts, meetings, books, reviews, letters and editorials; validation of instruments; articles written without academic peers; studies without full access to the original text	-

**Table 3 jfmk-10-00205-t003:** Classification of the general variables of the selected studies.

Author(s)/Ref.	N° Part	Sex	Age (Yrs) and Sports Level	Weight (kg)	Height (cm)	IMC (kg/m^2^)	Fat Mass (%)	Muscle Mass (kg */% **)
Elvira et al. [10]	12		20.3 ± 3.1	67.7 ± 9.0	171.2 ± 4.2		12.0 ± 1.2	45.8 ± 1.6 *
Moya-Ramón et al. [42]	12	♂	20.7 ± 4.8	70.4 ± 9.5	176.7 ± 5.6	22.5 ± 2.5	11.4 ± 3.7	84.1 ± 3.5 **
		♀	21.0 ± 4.2	64.1 ± 4.7	166.5 ± 6.4	23.1 ± 0.1	21.9 ± 1.8	74.2 ± 1.6 **
Daneshfar et al. [43]	12	♂	19.2 ± 3.5	68.5 ± 4.3	176.0 ± 0.06			
Mateo-March et al. [44]	14	♂	23.5 ± 0.5 (Elite)	77.1 ± 0.6	176.1 ± 0.3		11.1 ± 0.1	
			15.3 ± 0.5 (Cadet)	56.5 ± 0.4	165.7 ± 0.6		13.9 ± 0.2	
Robert et al. [45]	10		18.8 ± 3.7 (Elite)	68.4 ± 8.5	174.0 ± 9			
			19.8 ± 4.8	69.2 ± 11.7	170.0 ± 9			
Daneshfar et al. [23]	14		20.3 ± 1.5	70.2 ± 6.4	175.0 ± 0.05			
Petruolo et al. [46]	12	♂	18.9 ± 2.3	70.7 ± 5.0	174.0 ± 5.0		7.4 ± 1.6	
Gross and Gross [47]	12	♂	24.6 ± 2.6	72.2 ± 10.5	174.4 ± 8.0			
Peinado et al. [12]	12	♂	19.2 ± 3.4	72.4 ± 8.4	174.2 ± 5.3			
Hurst et al. [11]	60	♂	26.4 ± 8.4	75.3 ± 5.9	179.4 ± 7.2			
Rylands et al. [48]	6	♂	20.0 ± 2	68.0 ± 4	169.0 ± 5.0		10.0 ± 3	
Rylands et al. [49]	8		21.0 ± 2	69.0 ± 3	170.0± 6		10.0 ± 2	
Mateo-March et al. [25]	16		23.3 ± 0.9 (Elite)	77.7 ± 0.6	176.0 ± 0.4			
			15.8 ± 0.2 (Cadet)	56.4 ± 0.5	166.0 ± 0.3			
Louis et al. [50]	10	♂		75.4 ± 3.3	177.3 ± 9.7	24.2 ± 1.6		
		♀		61.2 ± 3.9	169.3 ± 6.9	21.4 ± 1.4		
Rylands et al. [17]	6	♂	18.0	67.1 ± 5.2	169.0 ± 8			
Mateo et al. [51]	11	♂	19.3 ± 2.1	72.2 ±10.37	172.2 ± 10.3	22.51 ± 3.2	11.5 ± 1.3	58.08 ± 1.17 *
Cowell et al. [20]	26	♂	26.0 ± 4					
		♀	22.0 ± 3					
Zabala et al. [9]	10		20.7 ± 1.4	77.9 ± 2.1	178.3 ± 2.1			
Mateo et al. [16]	9	♂	17.0 ± 1.41	71.9 ± 7.53	176.2 ± 4.61		16.4 ± 1.5	42.86 ± 4.29 *
Zabala et al. [19]	6		19.3 ± 2.1	75.1 ± 5.3	175.0 ± 0.05	21.1 ± 2.2	11.2 ± 2.1	
Zabala et al. [8]	9		19.4 ± 2.3	73.8 ± 9.9	174.5 ± 6.8			

**Note**: N° Part: number of participants; man: ♂; woman: ♀; Yrs: years; kg: kilograms; cm: centimeters; m: meters; %: percentage; blank: not applicable. * Evaluation of muscle mass in kg. ** Evaluation of muscle mass in %.

**Table 4 jfmk-10-00205-t004:** Classification of the methodological procedures of the studies.

Author(s)	Study’s Aim	Variables	Results	Instruments	Conclusions
Elvira et al. [10]	To describe the acceleration profile in BMX outputs and compare between NQ and Q plates.	SPM	Q showed no improvement in performance; force application time increased (*p* = 0.013, ES = 0.39), and dead point time reduced (*p* = 0.028, ES = −0.73).	The conventional Q-ring BMX chainring (Avent TR7, Avent Cycles, Murrieta, CA, USA), 44-tooth plate (XTR M970), stopwatch connected to two-timing ports (0.001 s of precision; SportMetrics, Picanya, Valencia, Spain), 6 g SignalFrame USBMe triaxial accelerometer (200 Hz sampling frequency; SportMetrics, Valencia, Spain).	Accelerometry and video analysis are useful in characterizing the ability to apply force in the first pedal strokes. The Q-ring noncircular pedaling system does not improve performance in BMX.
Moya-Ramón et al. [42]	To describe sprint characteristics in elite BMX cyclists and predict performance.	Maximum power	Significant correlations between performance and 1RM (*r* = 0.84, *R*^2^ = 0.65), maximum power in RST (*r* = 0.87, *R*^2^ = 0.78), and Wingate test (*r* = 0.68–0.77, *R*^2^ = 0.14–0.65).	Tanita Europe BV, Amsterdam, The Netherlands; Smith machine (Technogym Trading, Gambettola, Emi-lia-Romagna, Italy) registered with a linear encoder (Speed4lift, Madrid, Spain); Wattbike Pro (Wattbike Ltd., Nottingham, UK); ergometer; Chronojump (BoscoSystem, Barcelona, Spain).	Maximum power production (PPO): the PPO is the main predictor of performance in BMX, explaining 78% of the variability in full-lap performance.
Daneshfar et al. [43]	Evaluate physiological demands of BMX cycling on the track.	PWR, BLr	Back time correlated with average PWR; strong association between BLr and average return time.	Physical activity readiness questionnaire (PAR-Q); Seca 21 stadiometer, Birmingham, UK; a Seca Quadra 808 digital scale, Birmingham, UK; a Watt Bike Pro (Giant 2015, Nottingham UK); HR was monitored using a Garmin^TM^ (Garmin Olathe, KS, USA).	Both energy systems (aerobic and anaerobic) are associated with BMX performance. Coaches should consider these factors when conditioning cyclists.
Mateo-March et al. [44]	Compare BMX pedaling performance with circular and noncircular CR systems.	% PM	Elite: no significant differences between circular and noncircular CR; cadet: better performance with circular CR in % PM (*p* = 0.02).	Q-ring, conventional BMX bike (Redline Proline, Red83 line, Seattle, WA, USA); Polar RS800 HR monitor and 96-belt electrode transmitter T61); Rennen Design Group, Middleboro, MA, USA. The rear wheel was equipped with a GCog USA power meter.	The noncircular CR system does not improve performance in elite cyclists or cadets.
Robert et al. [45]	Evaluate the relationship between vertical jump and performance in BMX.	SJ, CMJ, DJ	Significant relationship between running time and jump height in SJ (*r* = −0.801, *p* = 0.017), CMJ (*r* = −0.798, *p* = 0.018), and DJ (*r* = −0.782, *p* = 0.022).	SECA©, Birmingham, UK (precision of 1 mm for height and 0.1 kg for weight); two photocells of the brand ARTEK^®^ PNP (Advanced Technical Lighting Projects, SL, Spain); Chronojump Boscosystem^®^, Barcelona, Spain (29.6 × 21 cm).	Jumping ability: there is a direct relationship between the best mark in BMX and jumping ability (SJ, CMJ, DJ), with higher absolute values in the elite group.
Daneshfar et al. [23]	Characterize energy production in BMX riders and investigate their role in running performance.	Power (W)	Significant correlation between stroke time and maximum relative power (r = −0.68, *p* < 0.01) and average power (r = −0.52, *p* < 0.01).	Digital scales Seca Quadra 808, Birmingham, UK; Stadiometer Seca 213, Birmingham, UK; standard electronic exit gate (Pro-Gate, Rockford, IL, USA) USA; NEOtm Swift Performance photocells, Queensland, Australia; the SRM Schoberer Rad Messetechnik training system was used; HR monitored with Garmin HR HRM-Dual™, USA.	Power data analysis helps cyclists apply strategies in energy production in certain parts of the track. Coaches must design programs based on race intensity and power output zones.
Petruolo et al. [46]	Investigate the physiological profile and demands of a simulated BMX career.	Vo2max; maximum power (W)	Vo2max: 55.7 ± 4.8 mL/min/kg; maximum power: 1498 ± 189 W; lactate and hydrogen ions increased significantly during series.	eEgometer (SRM Ergometer, Schoberer Rad Messtechnik, Jülich, North Rhine-Westphalia, Germany), portable blood lactate analyzer (Lactate Plus, NOVA Biomedical, Waltham, MA, USA), perceived exertion (RPE) measured using the Category-Ratio 10 Scale developed by Borg.	Elite cyclists show high anaerobic and neuromuscular qualities, while aerobic qualities are less relevant. The races induce metabolic alterations and fatigue but do not affect performance times in the series.
Gross and Gross [47]	Investigate if BMX cyclists exhibit similar Fv and torsion–cadence (Tc) characteristics in linear exercises and sport-specific tasks.	Fv and Tc	Divergences in the characteristics of Fv and Tc were found in ramp departures and flat ground sprints, suggesting the influence of specific technical skills.	- One-dimensional force plate (MLD Test EVO 2, SP Sport, Trins, Austria). *Muskelleistungsdiagnose 2010, version 5.2.0.6101, InfPro IT Solutions GmbH, Innsbruck, Austria. - Electronic timing doors (TC Timing System, Brower, Salt Lake City, UT, USA). - The power meter (Shimano DXR with SRM spider, SRM, Jülich, Germany) was modified with a gyroscope (Axiamo GmbH, Biel, Switzerland).	The ability to produce force at low speeds and maximum power are related, but there are discrepancies in high-speed parameters. Resistance training and specific technical aspects are essential for optimal balance in BMX.
Peinado et al. [12]	Test the effect of sodium bicarbonate ingestion on simulated BMX competition performance.	NaHCO_3_ was observed in the pH	Significant effect on pH, bicarbonate, and excess base (*p* < 0.05); unchanged in time, maximum rate, and time up to maximum rate (*p* > 0.05).	Blood gas analyzer (ABL 77, Radiometer, Copenhagen, Denmark) for pH; the blood lactate concentration ([La]) was analyzed using the enzymatic method (YSI 1500, Yellow Springs Instruments Co., Yellow Springs, OH, USA); SportMetrics cutoff photocells (sensitivity of 0.001 s); (Garmin International Inc., Olathe, KS, USA); Monitor 90 (Polar Electro, Kempele, Finland); portable gases Jaeger Oxycon Mobile (Erich Jaeger, Viasys Healthcare, Hoechberg, Germany).	Baking soda ingestion does not improve performance in a simulated BMX competition, although future studies should consider effects on recovery.
Hurst et al. [11]	Investigate the influence of helmets and collars on accelerations in young cyclists.	ROM, NB, WB	Significant differences in cervical flexion/extension, head movements, and translational accelerations between age groups.	- The collars (Atlas, Atlas Brace Technologies, Valencia, CA, USA). - Accelerometer with gyroscope (xPatch, X2 Biosystems, Seattle, WA, USA).	Head accelerations decrease with age and the use of collars. Strength work on the neck can help reduce these accelerations in young cyclists.
Rylands et al. [48]	Analyze optimal cadence for maximum power production in BMX cyclists.	Power (W)	Power (W) and maximum power (1105 ± 139 W) at 100 Rev./min; shorter time to produce energy at 120 Rev./min (2.5 ± 1.07 s).	SRM cycloergometer in isokinetic mode, standard 70 cm straight bar, Shimano SPD pedals, 175 cm connecting rod length, 50.2 kg·m^2^ inertial load, PowerTap hub-based powermeter system (Professional model, CycleOps, USA), photoelectric cells, Harpenden stadiometer, air displacement plethysmography (Bod Pod, Life Systems International, Charlotte, NC, USA).	Strength training and conditioning can maximize dynamic force production and select optimal transmission ratios.
Rylands et al. [49]	Determine variation in maximum power, torque, and power production time in BMX cyclists in the laboratory and the field.	Power (W)	Significant differences in maximum power and power production time between laboratory and field tests; not so in torque.	32-gauge cycloergometer; Shimano SPD pedals (Shimano Pedal Dynamics; Shimano, Inc., Osaka, Japan); SRMWin software (Version 4.3).	Laboratory data can underestimate performance compared to field data.
Mateo-March et al. [25]	Investigate the effect of noncircular chainrings on performance in the acceleration phase in BMX.	Output power	Elite group improved distance with Q-ring (+0.26 m, *p* = 0.02); cadet without significant improvement.	Q-ring; conventional BMX bike (Redline Proline, Redline, Seattle, WA, USA); Polar RS800 HR monitor and a T61 electrode transmitter belt; G-Cog power meter (Rennen Design Group, Middleboro, MA, USA).	The Q-ring system substantially improves the total distance in the initial sprint, although there are no differences in the power output, which may be sufficient to favor its use in competition.
Louis et al. [50]	Investigate physiological demands of Supercross BMX in elite athletes.	Vo2pico; Vo2max; blood lactate	High request for aerobic (Vo2peak: 94.3 ± 1.2% Vo2max) and anaerobic (blood lactate: 14.5 ± 4.5 mmol/L) glycolysis.	Ergocycle with electromagnetic brake (SRM, Schoberer. Rad Messtecnik, Jülich, Welldorf, Germany). Exhaled gases and oxygen consumption were immediately measured after each race using the K4b2 gas exchange system (COSMED, Rome, Italy). The analysis of electrolyte changes (anionic breach) and base excess was evaluated using a portable clinical blood analyzer (i-STAT™, Abbott, Princeton, NJ, USA).	The energy-dependent and oxygen-independent substrate routes are decisive for performance in BMX.
Rylands et al. [17]	Compare BMX speed production with other cycling disciplines.	Power (W)	Maximum power like other speed events (1539 ± 148 W); higher fatigue index in BMX.		Once maximum power is reached, pedaling cadence is the main factor for speed production.
Mateo et al. [51]	Analyze the effects of the BMX competition on anxiety perception and heart rate variability.	Cognitive anxiety; somatic and HR	Cognitive and somatic anxiety increased; correlation between anxiety and HR (*r* = 0.774, *p* < 0.05).	The CSAI-2R; Polar RS800 HR monitor configured in RR interval mode (Polar Electro, Kempele, Finland).	Precompetitive anxiety affects the dynamics of the heart rhythm, decreasing vagal control and increasing sympathetic activity. Repeated exposure to competitive stress can attenuate the perception of anxiety.
Cowell et al. [20]	Develop a greater understanding of Supercross bicycle motocross through notation analysis.	Video footage, movement patterns, and time spent pedaling, jumping, and “pumping”	Elite men took 39.62 ± 0.78 s to complete a track, using 30.45 ± 3.2 pedal strokes and dedicating 11.83 ± 1.11, 9.64 ± 1.79, and 17.05 ± 1.51 s to pedal, jump, and “pump”, respectively. The elite women took 40.95 ± 0.91 s to complete a track, using 33.65 ± 5.06 pedal strokes, and spent 14.40 ± 2.17, 6.28 ± 1.41, and 17.80 ± 1.83 s to pedal, jump, and “move and pump”.	Quicktime™ X (10.x) and VideoMotion© software (Version 2.0).	The dominant movement patterns investigated for take-off, landing, and pumping were hip extension (approximately 30 times per lap), knee extension (approximately 30 per leg per lap), and horizontal shoulder abduction and adduction (20 times per lap). Exercises that are specifically aimed at the extensors of the hips, knees, and ankles and the muscles responsible for the abduction and horizontal adduction of the shoulder are recommended.
Zabala et al. [9]	Examine the effect of sodium bicarbonate ingestion in consecutive sprints in BMX.	Variable acid–base pH	Significant differences in acid–base variables (*p* < 0.05); no significant differences in yield between trials.	Cycloergometry (Lode Excalibur, Groningen, The Netherlands); CMJ test; Ergojump, Rome, Italy; blood gases ABL77 (Radiometer, Copenhagen, Denmark); YSI 1500 (Yellow Springs Instruments Co., Yellow Springs, OH, USA).	Baking soda ingestion modifies the acid–base balance but does not improve performance on the Wingate test or the perception of preparation.
Mateo et al. [16]	Analyze the relationship between power production and race phases in BMX.	Power (W), FP	Maximum power influenced by the difficulty of the track; higher speed when pedaling is allowed.	The PowerTap SL 2.4 power meter, CycleOps/PowerTap SL, Madisson, WI, USA; Skywatch Xplorer 2 anemometer (JDC Electronics (Skywatch Xplorer), Yverdon les Bains, Switzerland).	The power and performance profile depend on the phases and techniques of the race, which are affected by the level of difficulty of the track.
Zabala et al. [19]	Determine the external feedback effect at the start time in BMX.	Departures	Significant reduction in onset time after treatment (1.264 ± 0.045 ms in PRE, 1.047 ± 0.019 ms in POS, 1.041 ± 0.021 ms in RET).	Machined door (Pro-Gate^®^); Panasonic NV DS68 video camera, Osaka, Japan.	Audiovisual feedback and cognitive training can significantly improve the execution of the beginning in BMX, reducing the time to develop the task.
Zabala et al. [8]	Examine the effect of sodium bicarbonate ingestion on simulated BMX rating performance.	Blood lactate	There were no significant differences in performance between sprints; blood lactate major after WT2 (*p* < 0.05).	Cycloergometry (Lode Excalibur, Groningen, The Netherlands); CMJ test; Ergojump, Rome, Italy; Dr. The Lange microphotometer (Berlin, Germany).	Performance and NaHCO_3_: it did not improve performance or perception of effort in a series of BMX ratings, although it improved the perception of preparation before each test.

**Note.** Q: Q-ring chainring; NQ: not-Q-ring chainring; SPM: statistical parametric mapping; RST: repeated sprint test; BLr: blood lactate response; PWR: peak power to weight ratio; % PM: pedaling percentage; SJ: Squat Jump; CMJ: Countermovement Jump; DJ: Drop Jump: Fv: force–velocity; Tc: torque–cadence; W: Watts; NaHCO_3_^−^: sodium bicarbonate; pH: hydrogen potential; HR: heart rate; Vo_2_max: maximum oxygen consumption; ES: effect sizes; NB: neck brace; WB: without brace; ROM: range of motion.

## Data Availability

The data confirming the results obtained are available through the corresponding authors.

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
