# Peer review of "A Systematic Review of Bicycle Motocross: Influence of Physiological, Biomechanical, Physical, and Psychological Indicators on Sport Performance"

_jfmk, 2025, doi:10.3390/jfmk10020205_

Round 1
Reviewer 1 Report
Comments and Suggestions for Authors
The authors present a systematic review that analyzes the multifactor determinants of BMX performance. Its greatest contribution is synthesizing and collating evidence from 21 studies relating to physiological, biomechanical, physical, and psychological variables influencing BMX performance.
The study is well-written and of good quality overall.
I would only suggest the following things:
- Include research questions that are being answered through this systematic review, so the readers are more focused on what this study will tell them
-
In some sections, citations are formatted as “[8.9]” while citations should use a comma separated style [8, 9]
I would suggest the paper is accepted for publication after minor revisions.
Comments on the Quality of English LanguagePlease fix instances of long-complex sentences:
-
- e.g. -> The study by Mateo-March et al. [25] evaluated the power in the specificity of the competition and demonstrated that the Pmax applied on average by the athlete was 85% of the Pmax that was reached in the laboratory; these values decreased at the beginning and beginning of the course, with values of 73% and 51%, respectively.
Go through the paper once again and check them.
Author Response
Letter to reviewer 1
Dear Reviewer,
Thank you for reviewing our manuscript. As a research group we value your effort and input.
We have followed your suggestions point by point to improve the manuscript quality, according to our possibilities. The changes have been made in the full text using the red color so that you can see them.
Thanks for your time. Once again, we thank you for your valuable contributions, which have helped to strengthen the document.
Comments 1. Include research questions that are being answered through this systematic review, so the readers are more focused on what this study will tell them
Response 1: The reviewer's comments were accepted. Questions were asked to share the results.
The questions asked were as follows:
4.1. What are the most important physiological variables?
4.2. Is there a relationship between physical capabilities and BMX performance?
4.3. What are the most studied biomechanical and technological variables related to BMX sport performance?
4.4. How do psychological variables influence BMX performance?
4.5. Skills and techniques in BMX?
Comments 2. In some sections, citations are formatted as “[8.9]” while citations should use a comma separated style [8, 9]
Response 2: Adjustments were made to all references throughout the document.
Comments 3. The study by Mateo-March et al. [25] evaluated the power in the specificity of the competition and demonstrated that the Pmax applied on average by the athlete was 85% of the Pmax that was reached in the laboratory; these values decreased at the beginning and beginning of the course, with values of 73% and 51%, respectively.
Response 3: The reviewer's comments were accepted. Each sentence of the document was reviewed for consistency.
The study by Mateo-March et al. [25] evaluated the power in the specificity of the competition and demonstrated that the Pmax applied on average by the athlete was 85% of the Pmax that was reached in the laboratory; these values decreased at the beginning with values of 73% and 51%, respectively.
Thank you for your positive feedback on our research. Your valuable suggestions greatly contributed to the improvement of our work.
Best regards

Reviewer 2 Report
Comments and Suggestions for Authors
Dear Authors,
This manuscript is very interesting, although it concerns a rather narrow group of readers due to the specificity of the topic. I believe that it was prepared reliably, using the methodology generally accepted for systematic reviews and meta-analysis. Table 4 in particular is evidence of a thorough analysis of the papers included in the review. I have no reservations about the content. I only have minor comments concerning mainly the abbreviations used.
line 460 NQ and Q plates –– the explanation of this abbreviation can be found below the table 4 but it should also be in the text when used for the first time, this note also applies to:
line463 - PS event
Line 464 SPM 1D
Line 498 CF, CF indices, CF Dynamics-(line 514), CF rates
Line 510 -CSAI-2R and FC
Additional questions and suggestions:
Are “QR noncircular pedaling system” (Line 469), and noncircular chainrings (Q rings) (line 474) the same terms?
I suggest that instead of the term “HR Dynamics”, the term heart rate variability be used, because it is generally accepted and recognized
Author Response
Letter to reviewer 2
Dear Reviewer,
Thank you for reviewing our manuscript. As a research group we value your effort and input.
We have followed your suggestions point by point to improve the manuscript quality, according to our possibilities. The changes have been made in the full text using the red color so that you can see them.
Thanks for your time. Once again, we thank you for your valuable contributions, which have helped to strengthen the document.
Comments 1. Line 460 NQ and Q plates –– the explanation of this abbreviation can be found below the table 4 but it should also be in the text when used for the first time, this note also applies to:
Response 1: The reviewer's comments were accepted. Added description of the acronym. Not-Q-Ring chainring (NQ) and Q-Ring chainring plates (Q).
Comments 2. Line 463 - PS event.
Response 2: The reviewer's comments were accepted. Added description of the acronym. pedal strokes (PS)
Comments 3. Line 464 SPM 1D
Response 3: The reviewer's comments were accepted. Added description of the acronym. statistical parametric mapping (SPM).
Comments 4. Line 498 CF, CF indices, CF Dynamics-(line 514), CF rates, Line 510 -CSAI-2R and FC
Response 4: The reviewer's comments were accepted. Explanations of each of the acronyms have been added the first time they are mentioned in the text.
Comments 5. Are “QR noncircular pedaling system” (Line 469), and noncircular chainrings (Q rings) (line 474) the same terms?
Response 5: The reviewer's comments were accepted.
Comments 6. I suggest that instead of the term “HR Dynamics”, the term heart rate variability be used, because it is generally accepted and recognized
Response 6: The reviewer's comments were accepted. The concept of heart rate variability was adopted throughout the text.
Thank you for your positive feedback on our research. Your valuable suggestions greatly contributed to the improvement of our work.
Best regards
